# Visualization, Interaction and Analysis of Heterogeneous Textbook Resources

**Christian Scheel** [1,*,†,‡], **Francesca Fallucchi** [1,2,‡]  **and Ernesto William De Luca** [1,2,†,‡]

1  Georg Eckert Institute for International Textbook Research Member of the Leibniz Association,
   38114 Braunschweig, Germany; f.fallucchi@unimarconi.it (F.F.); deluca@gei.de (E.W.D.L.)
2  Department of Innovation and Information Engineering, Guglielmo Marconi University, 00193 Rome, Italy
*  Correspondence: scheel@gei.de
†  Current address: Celler Straße 3, D-38114 Braunschweig, Germany
‡  These authors contributed equally to this work.

**Abstract:** Historically grown research projects, run by researchers with limited understanding of data sustainability, data reusability and standards, often lead to data silos. While the data are very valuable it can not be used by any service except the tool it was prepared for. Over the years, the number of such data graveyards will increase because new projects will always be designed from scratch. In this work we propose a Component Metadata Infrastructure (CMDI)-based approach for data rescue and data reuse, where data are retroactively joined into one repository minimizing the implementation effort of future research projects.

**Keywords:** textbook research; digital humanities; digital infrastructures; data analysis

## 1. Introduction

The availability of Big Data has boosted the rapidly emerging new research area of Digital Humanities (DH) [1,2] where computational methods have been developed and applied to support solving problems in the humanities and social sciences. In this context, the concept of Big Data has been revised and has another connotation, which regards the data being too big or complex to be analyzed manually by a close reading [3]. Educational media research suffers from this overwhelming availability of information and will likely get a boost by the DH, as it will be possible to analyze the sheer amount of historical textbooks on international level.

For instance, educational media research investigates sanctioned knowledge by comparing textbooks and identifying modified, missing or added information. Such modifications can have a big impact for the formation of the young generation. Hence, textbooks are also gaining importance in the historical research [4]. Because there are millions of digitized textbook pages available, researcher's search for "popular knowledge", as it reflects views of the world, thought flows and desired knowledge has to be supported by digital research tools. To be able to work with processed data, the digital research tools rely on digitization efforts and people who make implicit knowledge explicit by describing said resources. In this direction, we developed Toolbox [5] a complete suite for analysis in Digital Humanities. The system offers functionalities that allow text digitalization, Optical Character Recognition(OCR), language recognition, digital libraries management, topic modeling, word counting, text enrichment and specific reporting elements, all in a flexible and highly scalable architecture.

From a digital research point of view, data derived from textbooks and educational media provide interesting challenges. First, processed data were often not meant to be reused. Hence, the data are hard to retrieve and has to be processed again to be of any value for the research community. Second, if processed data are available, they are syntactically heterogeneous (text, images, videos, structured

data in different formats, such as XML, JSON, CSV and RDF), they are described in different metadata standards and often written (or cataloged) in different languages, without any use of standards or controlled vocabulary. Third, the data are semantically rich, covering different "views of the world" taken from textbooks of different countries and epochs. Lastly, the data are implicitly interlinked across different data sources, but only accessible by individual and often outdated interfaces.

The separate storage of data silos is not helpful if humanistic researchers want to deal with such data and address semantically complex problems or interesting methodological problems. As a non-university research institution, the Georg Eckert Institute for the International Textbook Research (GEI) (http://www.gei.de/home.html).

Conducts and facilitates fundamental research into textbooks and educational media primarily driven by history and cultural studies. For this purpose, the GEI provides research infrastructures such as its renowned research library and various dedicated digital information services. Hence, the institute develops and manages both digital and social research infrastructures. As such, the GEI realizes a unique position in the international field of textbook research. In the digital humanities, the investigation of research questions is supported by a range of increasingly sophisticated digital methods such as automatic image and text analysis, linguistic text annotation, or data visualization. Digital tools and services combined with the increasing amount of resources available through digital libraries such as the German Digital Library, the Deutsches Textarchiv, Europeana and research infrastructures such as Common Language Resources and Technology Infrastructure(CLARIN) or Digital Research Infrastructure for the Arts and Humanities (DARIAH) provide digital support for textbook analysis.

Analogous to the work done in [6] we identified three generations of portals that develop historically grown and individually processed research projects. First, the research focus in semantic portal development was on data harmonization, aggregation, search and browsing ("first generation systems"). At the moment, the rise of Digital Humanities research has started to shift the focus to providing the user with integrated tools for solving research problems in interactive ways ("second generation systems"). The future portals not only provide tools to solve problems, but can also be used for finding research problems in the first place, for addressing them and even for solving them automatically by themselves under the constraints set by the humanities; however, to reach or even think about the possibilities of such "third generation system", some challenges like semantic interoperability and data aggregation have to be approached first.

In order to being able to embed institute's data into these resources, it has to be separated from existing historically grown research tools, to be joined in a single repository. Research projects, tailored for specific research questions, often result in graphical interfaces only usable for satisfying one given information need. Nevertheless, the underlying data are not limited to such use cases and could often also be used for searching, visualizing and exploring data. In this work, we show how overlaps and missing overlaps of these data silos can be disclosed with the Component Metadata Infrastructure (CMDI) [7] approach in order to retroactively disclose planning deficits in each project. Generalizing these shortcomings helps projects to be more focused on data reuse, user group multilingualism, the provision of standardized interfaces and the use of unified architectures and tools.

After describing the problem in detail and giving an overview about the lessons learned from visualization, data exploration and interactive data approaches, we show how to overcome the dispersion produced by data silos, a multitude of metadata formats and outdated tools using the CMDI, suggested by CLARIN. CMDI can help to not only emphasize the common characteristics in the data, but also keep the differences. Concluding, we show that the visualization, data exploration and interactive data approaches can be applied to the newly created repository, gaining additional research value from the newly known interconnections between the formerly separated data.

## 2. Problem Description

In the recent past, the Georg Eckert Institute has generated many data silos whose origins lie in historically grown and individually processed research projects. The data available in search indices or databases are fundamentally different, but have many common characteristics (such as title, persons, year and link to resource). Because the institute prescribes the research direction, the data from the research projects are thematically related, which is reflected not only in the common characteristics but also in their characteristic values. The separate storage of data silos is not desirable because, firstly, data is kept twice and, secondly, no project can benefit from the other.

In the following, the data, their similarities and their significance for data harmonization and interconnection are described, followed by data approaches (visualization, exploration and interaction), which can be observed on this data, to raise a common understanding of their potential benefits for a harmonized data repository.

### 2.1. Recording Characteristics and Characteristic Values

We started to explore each project's underlying data, in order to merge them and to get rid of data silos. Initial investigations had shown that the data structure was always very flat, even when complex objects were described. Whenever certain characteristics were present in most projects, but could not be satisfied by another project's data sources, the question was how and where to extract or substitute it from other projects' underlying data. We organized different workshops and analyzed project documentations together with the experts of the research field and with the users of the corresponding research tools. We learned that the observed differences between the common characteristic expressions resulted from missing knowledge about former and current projects. Hence, having common vocabulary, coming from standards or standard files, has never been an option. For merging projects' data and applying standards or standard files, we analyzed the following twelve project's resources for their characteristics:

- edu.docs (http://www.gei.de/en/departments/digital-information-and-research-infrastructures/edumeres-the-virtual-network-for-international-textbook-research/edudocs-publications-from-educational-media-research.html) (202 resources)
- edu.reviews (http://www.gei.de/en/departments/digital-information-and-research-infrastructures/edumeres-the-virtual-network-for-international-textbook-research/edureviews-the-platform-for-textbook-reviews.html) (371 resources)
- edu.data (http://www.gei.de/en/departments/digital-information-and-research-infrastructures/edumeres-the-virtual-network-for-international-textbook-research/edudata-textbook-systems-worldwide.html) (2796 resources)
- edu.news (http://www.gei.de/en/departments/digital-information-and-research-infrastructures/edumeres-the-virtual-network-for-international-textbook-research/edunews-the-latest-from-educational-media-research.html) (4064 resources)
- Curricula\Workstation (https://curricula-workstation.edumeres.net/en/curricula/) (7687 resources)
- K10plus (https://www.bszgbv.de/services/k10plus/) (search index of the library; 183,295 resources)
- GEI.de (http://www.gei.de) (the institute's website; 546 resources)
- GEI|DZS (https://gei-dzs.edumeres.net/en/search/) (2641 resources)
- WorldViews (http://worldviews.gei.de) (57 resources)
- GEI Digital (http://gei-digital.gei.de/viewer/) (5200 resources)
- Pruzzenland (https://www.pruzzenland.eu) (116 resources)
- Zwischentöne (https://www.zwischentoene.info/themen.html) (461 resources)

Below, we report an analysis of the most important bibliographic metadata used to record resource information in the different projects. When preparing the harmonization of data from different data sources, it is important to focus on the similarities of these resources, in order to not getting overwhelmed by individual differences. Additionally, these similarities are most likely the data which can interconnect the resources.

We formalize a bibliographic dataset ($D$) as follows. $D$ is a set of 8-tuple $d \in D = (id; url; t; p; c; T; s; l)$ where:

**id** are unique identifiers of the resource or to other resources.
**url** is a link to the resource.
**t** is the title of the resource.
**p** is the published/publisher of the resource.
**c** is the created/changed information of resource.
**T** represents the topics of the resource, a set of three resources $T = k, sa, dt$ where

　　**k** are the keywords or tags.
　　**sa** are the subject areas.
　　**dt** are places or descriptive terms.

**s** is the information related to level of education, school type, country of use or subject of the resource.
**l** is the language of the resource.

### 2.1.1. Identifiers

The most straight forward way of data harmonization is looking for the same identifiers within different resources and hence, identifying two descriptions $d_1$ and $d_2$ of the same resource or resources which are linked to each other. Although there is often a field "id" in the data, this attribute is not necessarily the data to look for, because it often just separates this resource from other resources of the same source. Talking with experts about exemplary resources will provide knowledge about fields which contain identifiers.

### 2.1.2. URL

When merging data from different sources, the URL should be used to reference the original data. The URL describes a fixed web address, which can be used to view an entry in the corresponding project. Accordingly, all URLs are different and cannot be limited by a prescribed vocabulary. We observed indirect URLs, where links led to a descriptive overview pages, generated by the containing projects. For data harmonization, these URLs had no value, because the information presented on these pages was already part of $d$. Within 7 of the 12 projects there was at least one link that led to the original resource. With the remaining five projects, the URL could be assembled with the help of static character strings and available information (e.g., from identifiers). The total coverage of this metadata was 99.83%.

### 2.1.3. Title

Titles are a very short textual description of an entry and are often combined with the URL to create a human readable link to the original resource. Intuitively one would assume that every entry in every project has a title. However, this is only the case for 99.62% of the resources. In all but four projects, there was a complete title assignment. Further investigations have shown that some of these documents were not missing the title in the original data source, but must have been lost when preparing the data for searching and presentation for the project's interface. For some resources, such as maps, a title was not always necessary.

### 2.1.4. Published, Publisher

Knowing when and by whom an entry was published is an important feature. Three of the twelve projects did not have this feature at all. This includes the institute's website, Pruzzenland and edu.data.

The information on the publisher was also missing for two other projects: edu.news and Zwischentöne. In case of missing publisher information, the publisher often was the institute itself. The total coverage of "published" is 92.31% and that of "publisher" 91.33%.

### 2.1.5. Created, Changed

Since the project's individual search indices have never been deleted and providing this service ever since, we were able to gain two pieces of information that are of great value for a common representation. The values "created" and "changed" managed by the search index were not found in the underlying databases. However, they describe very well and independently from the publication date when an entry was added to the corresponding project. It can also be considered as a substitute for publication date if this information is missing. Information on "changed" was available in 11 of 12 projects and on "created" in 10 of 12. The total coverage of "changed" was 94.64% and that of "created" 10.89%, because the research library resources are missing this information.

### 2.1.6. Topic

By topic we mean keywords, subject areas, places or descriptive terms. Even if they were not necessarily a descriptive topic term in the original project, the total coverage is 95.26%. Interestingly, it was news related project (edu.news) where such descriptive information is missing. This shows retroactively an error with the conception of this project, because keywords and geographical information are common information in news articles. Fortunately, the keywords and topics often have been linked with external knowledge bases (e.g., GND).

### 2.1.7. Level of Education, School Type, Country of Use, School Subject

Because it was important for educational media research, but is not part of traditional cataloging, the institute decided to establish a classification for textbook characteristics. The research, the textbook collection and hence the local classification scheme of the Georg Eckert Institute are primarily focused on educational sciences, history, geography, political science and religious sciences [8–10]. As these characteristics are specific characteristics of textbooks and related media, this information had the greatest overlap between the projects. However, recent projects have shown the need to map "level of education" and "school subject" into the UNESCO International Standard Classification of Education (ISCED) to be able to cover international educational media [4].

### 2.1.8. Language

Information about the language of the entries were often given implicitly, like when the whole data source was written in one language. The language in which the entries were written is unknown in half of the projects.

### *2.2. Data Approaches*

Research projects in our increasingly data- and knowledge-driven world are dependent on applications that build upon the capability to transparently fetch heterogeneous yet implicitly connected data from multiple, independent sources. Even though, all projects have been driven by the respective research goals, the resulting tools generally show how research could benefit from data-driven visualization, exploration and interaction approaches. The data inspection described in Section 2.1 made it obvious that there would be no short term solution for harmonizing underlying data of the projects, so that the projects could switch to the new data repository. Instead it revealed the long-term need to research and develop new projects that could interact together with the very large amounts of complex, interlinked, multi-dimensional data, throughout its management cycle, from generation to capture, enrichment in use and reuse and sharing beyond its original project context. Furthermore, the possibility of traversing links defined within a dataset or across

independently-curated datasets should be an essential feature of the resulting tools and thus to ensure the benefits for the Linked Data (LD) [11] community.

In the following, we further describe the reuse and reusability of the products of the different projects analyzing the benefit from data-driven visualization, exploration and interaction approaches in more details.

### 2.2.1. Visualizing Data

The design of user interfaces for LD, and more specifically interfaces that represent the data visually, play a central role in this respect [12–14]. Well-designed visualizations harness the powerful capabilities of the human perceptual system, providing users with rich representations of the data. Dadzie and Pietriga illustrate in [15] the design and construction of intuitive means for end-users to obtain new insight and gather more knowledge. As a cultural institution, the GEI digitizes and interlinks its collections providing new opportunities of navigation and search. However, it is comprehensive that the data are sparse, complex and difficult to interact with, so that a good design and support of the systems is indispensable. Moreover, it is difficult to grasp their distribution and extent across a variety of dimensions.

An important promise in connection with the digitization efforts of many institutions of cultural heritage is increased access to our cultural heritage [16]. Aggregators, such as the Digital Public Library of America and Europeana expand this ambition by integrating contents from many collecting institutions so as to let people search through millions of artifacts of varied origins. Due to the size and diversity of such composite collections, it can be difficult to get a sense of the patterns and relationships hidden in the aggregated data and the overall extent of the collection [17]. Therefore, new ways of interaction and visualization possibilities can help in finding relevant information more easily than before [18,19]. We developed different tools and visualizations for accessing educational media data in various project.

An example is given by the platform GEI-Digital (http://gei-digital.gei.de/viewer/), which is a first generation system that provides more than 4300 digitized historical German textbooks in the fields of history, geography and politics, including structural data (e.g., table of contents and table of figures) and OCR processed text from more than one million pages. Both textbooks from the Georg Eckert Institute and textbooks from other partner libraries were digitized and integrated. GEI-Digital aggregates the entire collection of German textbooks until 1918. In the course of digitization, a total of 250,000 metadata were recorded, whereby the indexing follows the specific needs of textbook research. Thus, in addition to information about the publisher and year of publication, subjects and grades were recorded as meta data. However this tool does not provide any visually appealing information, except from the presentation of the scanned textbook pages and figures, which offers researchers the opportunity to print sections and work directly on these copies [20].

To overcome this deficit, the prototypical visualizations of "GEI-Digital visualized" (http://gei-digital.gei.de/visualized/) have been developed in cooperation with the Potsdam University of Applied Sciences in the Urban Complexity Lab as part of a research commission from the Georg Eckert Institute for International Textbook Research. Through the visualization of the metadata and interactive combination possibilities, developments on the historical textbook market with its actors and products can be made visible. This tool illustrates the prerequisites and possibilities of data visualization, while being limited to only data coming from GEI-Digital [21]. Letting researchers use this tool and observing their interaction, we analyzed the added value given by data visualizations in combination with library content, on the one hand, and the research purposes on the other (see Figure 1).

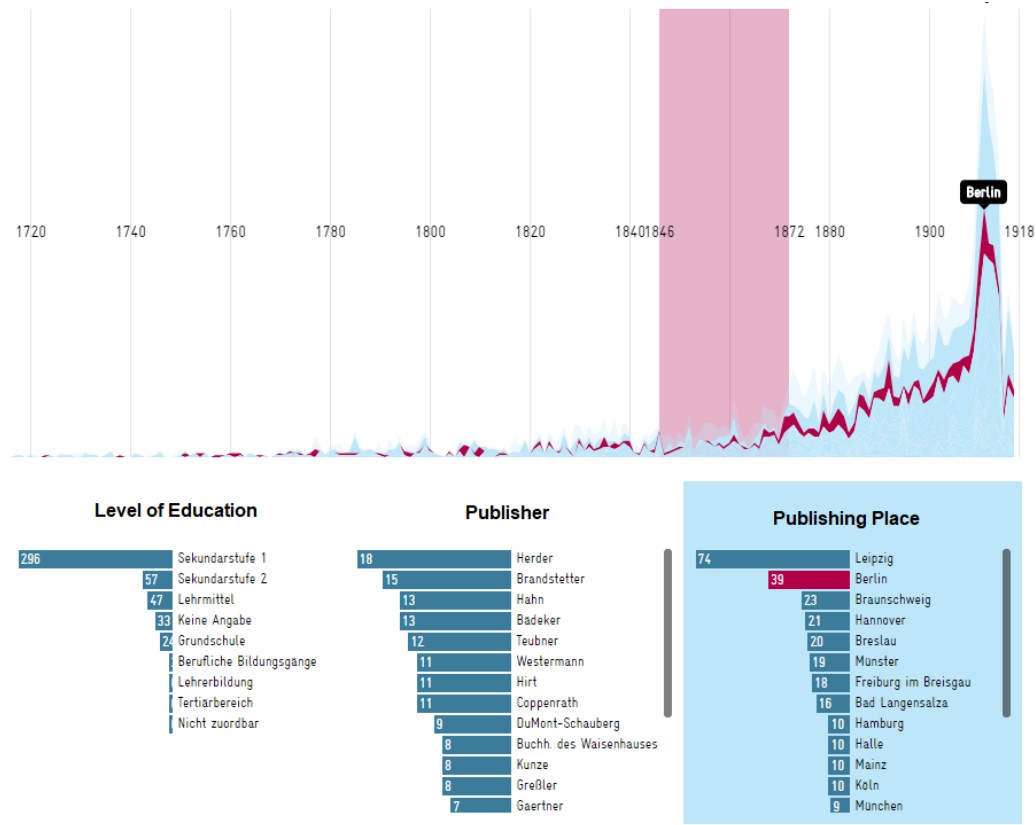

**Figure 1.** Screenshot of the Georg Eckert Institute for the International Textbook Research (GEI)-Digital-Visualized tool.

2.2.2. Data Exploration

Within the "Children and their world" Explorer, we implemented a second generation tool, which shows how texts, included in the corpus, can be exported and used in other DH tools for further detailed analysis (see Figure 2). Researchers can work with a set of texts and look for ways to reveal structural patterns in their sources, which were, until now, impossible to analyze within a classical hermeneutical way. This interdisciplinary DH project deals with world knowledge of the 19th-century reading books and children's books. The digital information (a sub corpus of the GEI-Digital textbook collection combined with the Hobrecker collection [22], a children's book collection) has been combined to implement specific tools for semantic search and statistical text analysis, which can support researchers to better formulate their research questions and to support the serendipity effect, which can be given by the use of digital tools. To this end, approximately 4300 digitized and curated 19th-century historical textbooks have been annotated at the page level using topic modeling and automatic enrichment with additional meta data. These extensions enable a free browsing possibility and a complex content and meta data driven search process on textbooks. For supporting the research goals of this project, a sub set of the books were manually annotated by the supposed target gender (male, female, both or unknown) or the targeted religious confession. The International TextbookCat research instrument (see Figures 3 and 4) does not only provide a welcome extension to the library OPAC system, but also is a discovery tool that dramatically improves the (re)search possibilities within underlying textbook collections. In contrast to the content driven "Children and their world" Explorer, which is dependent on the digitization process, the International TextbookCat is solely based on metadata and hence provides access to much more textbooks. It employs the local classification system (see Section 2.1.7) in order to categorize textbooks according to applicable country, education level and subject. Additional categories of federal state and school type are provided for German textbooks. The project extends the textbook collection with the inventories

of international partners, combining the textbook databases of three institutions: the Georg Eckert Institute (165,231 resources), the University of Turin (25,661 resources) and the National Distance Education University in Spain (66,556 resources), in order to create a joint reference tool [23]. Workflows and system architecture have been developed that in the long-term will enable further institutions to participate with relatively little effort on their part. An additional functionality is given in the statistics view. Diagrams illustrate features and compositions of the collection or currently selected set (see Figure 4), which on its own can be seen as an visualization approach. Researchers can use this feature for the development or verification of their hypothesis and research questions.

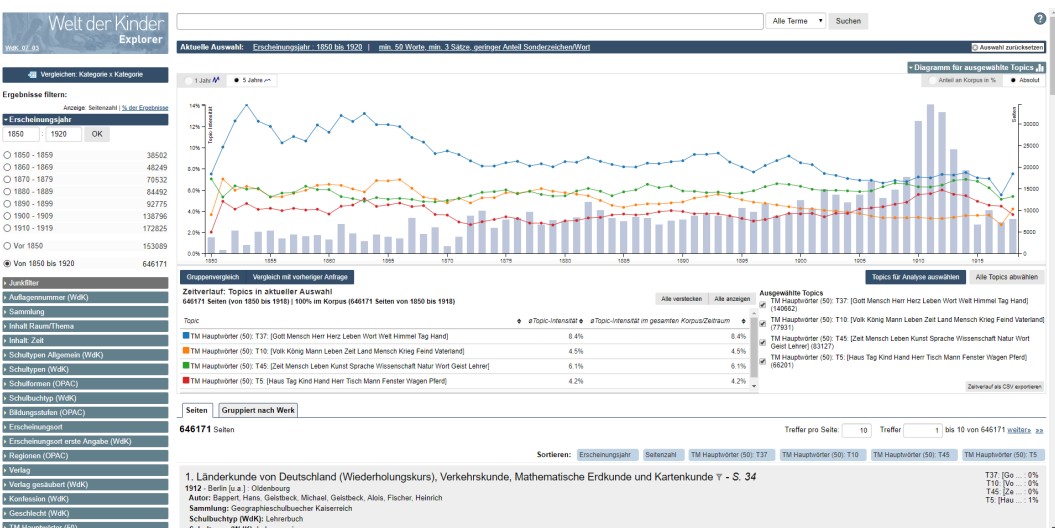

**Figure 2.** Screenshot of the Digital Humanities Tool "Children and their world" Explorer.

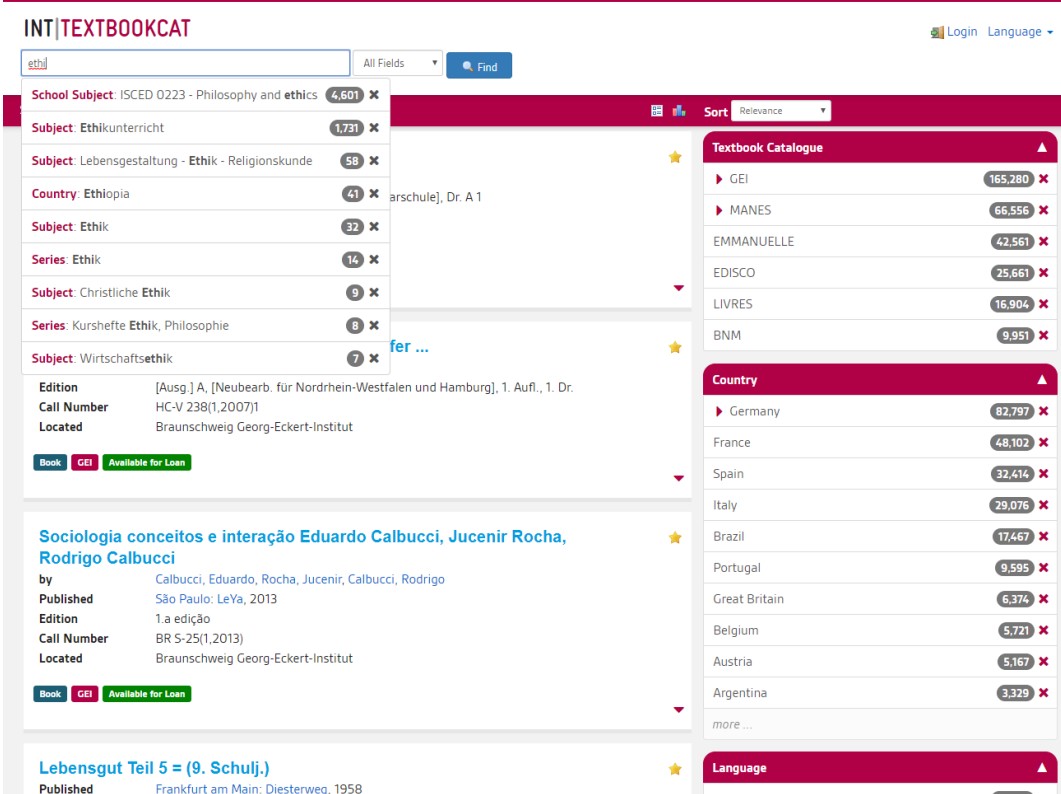

**Figure 3.** Screenshot of the International TexbookCat Research Tool.

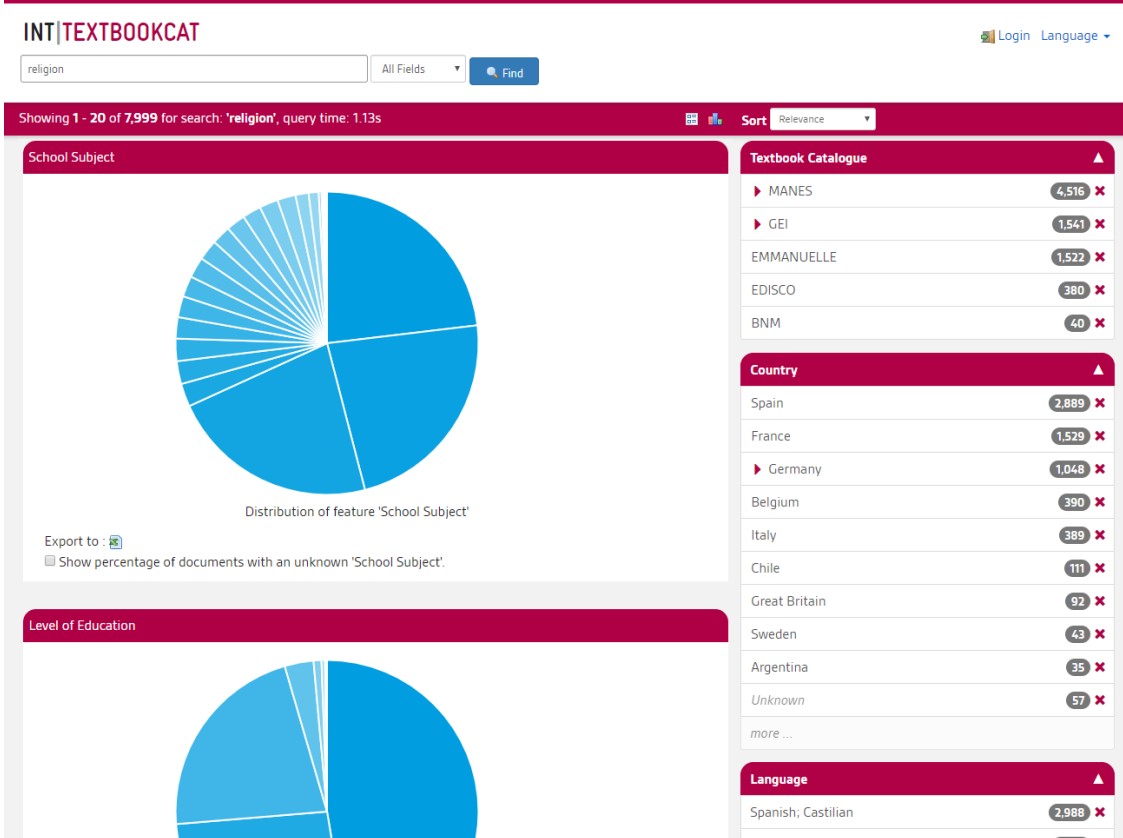

**Figure 4.** Presentation of the statistics given a query in the International TexbookCat Research Tool.

### 2.2.3. Data Interaction

The presented approaches encourage the user to interact with their underlying data in order to examine research questions or just in hope of the serendipity effects which could lead to new hypotheses. However, this interaction has no effect on the data itself. None of the examined projects used interaction data to improve itself. However, some projects encouraged the researcher to interact with the institute in order to create reviews, recommend new textbooks to purchase, prioritize books in the digitalization queue, etc.

Recently, we researched about the most desired features for textbook annotation tools and then created SemKoS, an annotation tool prototype, which supports data interaction and creation, based on digitized textbooks. In order to maximize the acceptance of the tool, we did a survey in which we found out what researchers expect from such a tool [20]. As a direct consequence, it was decided that annotations should be made directly on the scanned book pages and not on the corresponding recognized texts, as this supports a working method similar to the traditional work on a book. As it can be seen in Figure 5, text (representing entities) can be linked to a knowledge base, resulting in a better contextual understanding of the textbooks and the development of better data approaches in the future.

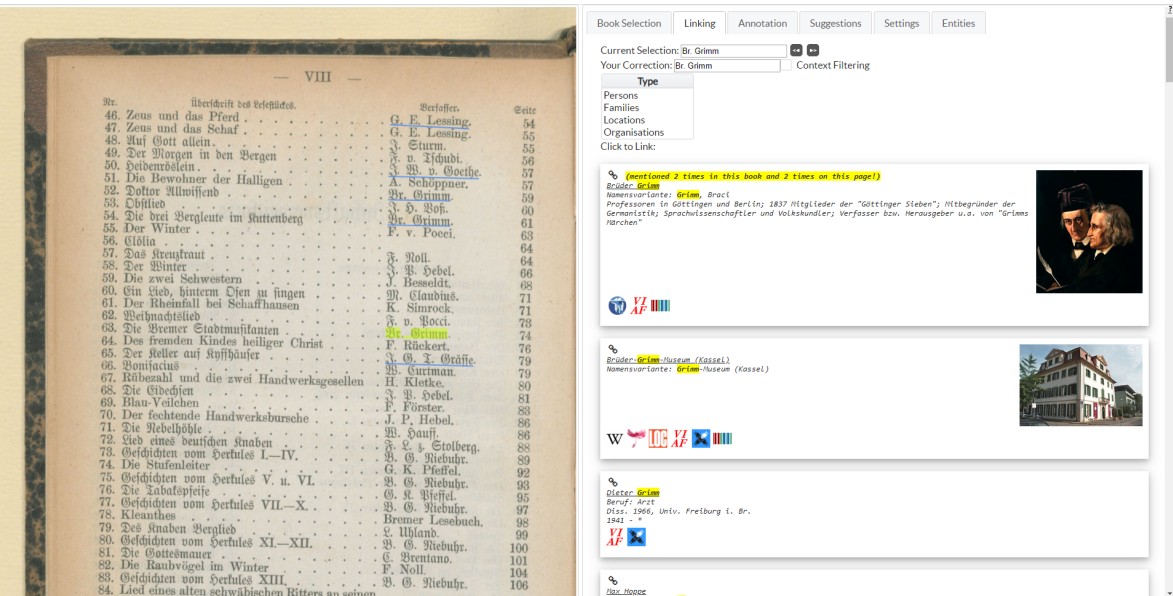

**Figure 5.** Screenshot of the SemKoS entity linking tool.

## 3. Creating a Middleware for Continuously Accessing and Joining Data

Developments in the direction of data integration and homogenization has been made within the project WorldViews [24] with its aim of establishing a middleware to facilitate data storing and reuse within the GEI context and beyond that. The project data, being stored in a standard format and accessible through standard interfaces and exchange protocols, will serve as a use case to test the data infrastructure's improvement to facilitate the data's long term sustainability and reuse. To intensify the connection to the Cultural Heritage World (like Deutsche Digitale Bibliothek or Europeana), creating a theoretical knowledge model, covering all types of resources, was essential. The data were found to be in various formats and stem from international and multilingual sources (see Section 2.1). Furthermore, in most project sources, the data were not static. Hence, the middleware had to be able to continuously access and join the data, where joining included cleaning up, mapping to a common representation and representing it in CMDI.

### 3.1. Accessing the Data

Since the research projects were driven by historical focused research questions, ignorant of the possibilities of later disclosure and reuse of the data, the data structure has been very neglected. Access to the data could be obtained in three ways:

1.  Browsing a web based user interface. The projects often offer a search, where the resources data is presented in a "detailed view".
2.  Analyzing the internal database of the architectures that make the web presence possible.
3.  Analyzing the search indices generated by the provided search functionality.

In an attempt to identify commonalities between the research projects, researchers from the institute took the path (1). In particular, the search masks were examined here, since its drop-down lists often showed the complete assignment of a property (controlled vocabulary). In addition, it was always tracked back where this information originally came from. At the same time, computer scientists tackled (2) and (3), where (2) was too time consuming due to the multitude of different architectures and the associated unmanageable variety of data. The data of the (separately kept) search indices (3) were comparatively easy to access, since they were kept in the same architecture (Solr) (https://lucene.apache.org/solr/) and hence, could be automatically accessed via interfaces.

### 3.2. Mapping the Data to a Common Representation

Defining the mapping of data available in the institute's digital information systems and services was the most time consuming part of data harmonization, because every single feature expression needed a representation in CMDI. In [25–27] we described our previous works on CMDIfication process for GEI textbook resources. Often researchers and users were needed to link each expression from the data to the common representation, because these persons knew the real meaning of expressions and would not make assumptions. This process led to a set of mapping rules (like "map language:'English' to iso_639_3:'eng' "), which could always be adjusted, extended and applied again, because mapping tools do not manipulate the original data, but the representations in CMDI.

### 3.3. Application of CMDI Profiles

When developing CMDI, CLARIN assumed that metadata for language resources and tools existed in a variety of formats, the descriptions of which contained specific information for a particular research community. The data within the research projects of the GEI (see Section 2.1), which have been tailored for the closed educational media research community, supports this assumption. Thus, as in CMDI, components can be grouped into prefabricated profiles. The component registry then serves to share components and profiles across the research projects and eventually to make them available to the research community.

### 3.4. Proof of Concepts

A variety of independent digital services and projects have been implemented in the past, so that researchers which were interested in cross-search possibilities had to find a way themselves to get the most relevant information related to their research questions, if they wanted to use different services. Moreover, researchers who were not familiar with the institute's services and data had no chance of knowing whether it would be worthwhile to learn how to use the tools.

To tackle this issue, the first application using the newly created joint repository was an institute wide search engine. Having the data in one place and knowing its origin, the task of creating this search engine was just configuring a search indexer and designing the user interface. Researchers are now supported in performing cross-research questions, analyzing the data from different perspectives (as shown in Figure 6 (http://search.gei.de) ) and analyzing found results in their original services in detail. The flow of data from the search engine point of view can be summarized as follows.

1. Project's data is accessed and retrieved continuously by the middleware.
2. Where applicable, the data are then mapped into standards, controlled vocabularies or codes.
3. The resources are then represented in CMDI and stored into a repository.
4. Repository's data are accessed and retrieved continuously by the indexer.
5. The indexer transforms the CMDI representation into a index document representation and stores it in the search index.
6. The index is accessed by the discovery tool (VuFind), which presents results to the user.
7. The codes are translated into corresponding terms of the selected/detected language.

While these steps look overly complicated, only steps (4) to (6) had to be implemented to create the search engine. In fact, in the future, step (4) should be provided by the repository's Application Programming Interface (API), where XSLT(eXtensible Stylesheet Language Transformations) could transform CMDI into other representations.

A second proof of concept did not result in a tool yet, but in supporting a research question. The newly created repository contained and linked data from GEI | DZS, an interface for filtering for approved textbooks, from the Curricula Workstation, a tool for searching for curricula, and textbooks from the library (via Findex). There was the need to filter textbooks for specific criteria, like school

subject, year of admission and level of education, to link the corresponding curricula to the textbooks. Each curriculum describes the wanted knowledge and if a textbook has been approved, we know for sure that this knowledge is in the book. Showing, that linking actual curricula to textbooks enables a variety of new text based approaches. The connection between these three data sources could not be made before, because the data was never meant to be used in other services than they were made for.

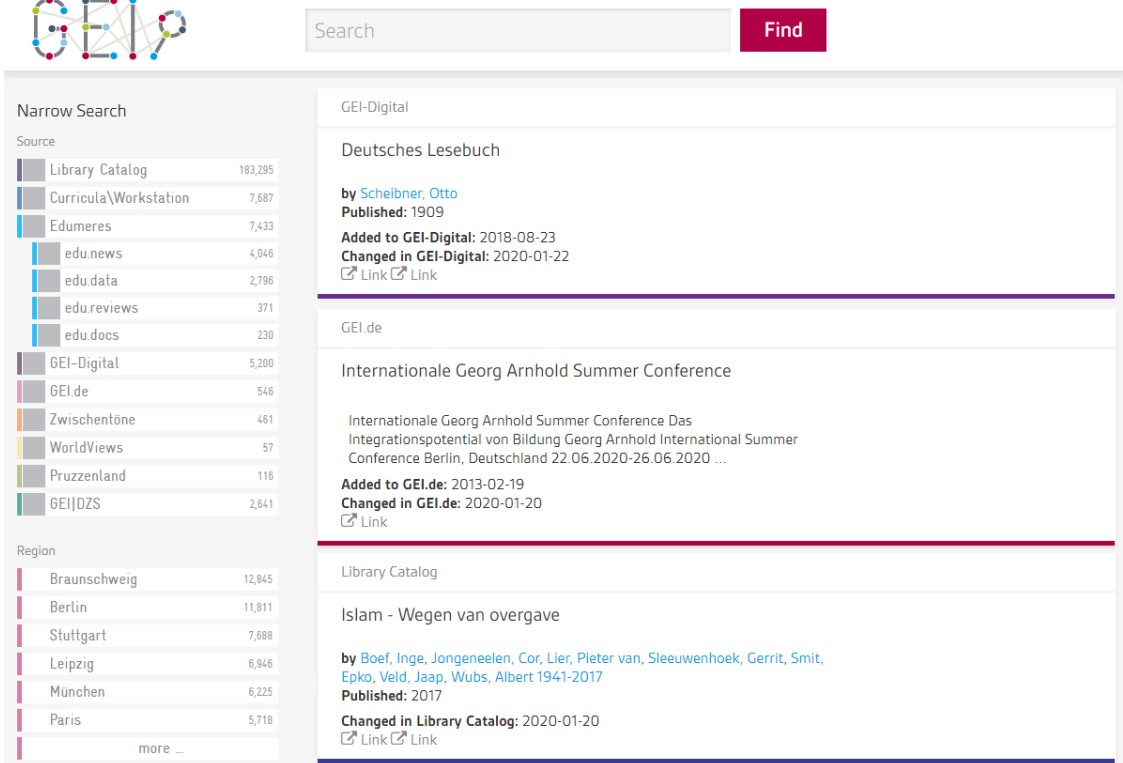

**Figure 6.** Meta search of data collections.

## 4. Discussion

In the recent past, the GEI has generated many data silos whose origins lie in historically grown and individually processed research projects. To get rid of these data silos, there was the need to harmonize all project's underlying data. Hence, we started collecting projects' documentations and investigated all options.

After analyzing the data, projects, tools and services of the GEI, we became aware of the great potentials. Not only did we find valuable data, but also tools and services for visualization, exploration and interaction. Even though the tools were designed for different purposes, the general ideas of these applications could be expanded to all the data. For instance, the technology for visually browsing through textbooks (like in GEI-Digital visualized) could be reapplied to visually browsing through textbook admissions or curricula.

The institute approached this undertaking from two perspectives. The researchers (projects' users) reviewed the user interfaces to conclude the data structure and tried to track the data back to its original source. The computer scientists performed a technical approach by analyzing the data in the back-end. While gathering more and more information and understanding where, why and how underlying data were stored, we had to solve new issues on the way. It has been shown that the technical approach can also reveal missing features within a source, while the manual investigation of the source only concluded that this feature exists. Conversely, the manual approach could not detect the existence of properties if they occurred infrequently. From the title of a given document, we could see that the transfer of data from the databases to the search indices did not always have to be complete. Software or planning errors in the corresponding base architectures can lead to more information on

the entries being available than can be found in the search index. Missing fields for publisher and corresponding publication date showed that one should also include implicitly given characteristics in the metadata when planning services, because these can be relevant with a subsequent use. The search indices contained values that could not be found in the databases and the user interface of the projects. Information such as when an entry came into a project or when it was last edited is required if someone wants to know what has changed in the project or if a project is still being managed. When planning new projects, such values should be considered as database entries. A deleted index can be rebuilt at any time, but this information could no longer be reproduced. The lack of information about educational level, school type, country of assignment, subject in edu.reviews' offer is a clear call for the reuse and linking of data, because exactly this information about the reviewed textbooks is contained in the library catalog. Even if the language of the entries is unknown in half of the projects, the technology has now reached the point where the language can be reliably determined and added. This example is representative of many metadata that can be derived from other sources or supplemented in order to make the existing data as complete as possible. Metadata fields such as keywords, subject areas and locations can often also be re-used as general topics. Such a field would be comparable with the GND keywords, which are equally diverse. This means that a field is created here which does not necessarily have to occur in any project. Here the data-driven approach was advantageous, because all topics could be assigned an ID, which links the keywords with the GND. The use of IDs instead of natural language entries also promotes multilingualism, since linked data is often translated into different languages. A decisive advantage when investigating the interface was that the experts always asked themselves: "Where does this data come from?" Even though the indices provide a good overview, they also showed that data were manipulated or lost on the way to the index. Knowing which source they were fed from is indispensable for setting up component registry. The evaluation showed that it was useful to analyze the data in parallel by experts who were familiar with the database and the projects and computer scientists who combined similarly filled index fields pragmatically to create the basis for a common database. The approach of the subject scientists led to detailed investigations of the characteristic values of meaningful characteristics, while the approach of the computer scientists revealed common characteristics. Both approaches complemented each other to enable the generation of CMDI profiles and the transfer of data to component registry.

This work and the succeeding work of representing and linking the analyzed data sources into CMDI and storing it in a repository, enables the implementation of new tools in the future. These tools may be similar to the existing tools, but then with the underlying data always being up to date. The tools may then cover all the data instead of the part they were designed for (compare to the GEI-Digital-Visualized tool). The interconnections may lead to new tools, or, to begin with, link one service with another. Linking also enables the derivation of additional textbook attributes. For instance, if we know the knowledge described in curricula and know the approved textbooks for these curricula, then we know the containing knowledge of the textbooks without any digitization effort.

## 5. Conclusions

In this work, we identified various reasons to join the data behind digital services. We illustrated the challenges, but also the opportunities of harmonizing such data retroactively, having a single point of access, using the Component Metadata Infrastructure (CMDI), which was especially designed for such an undertaking. To show the many advantages of such data repository, we implemented a prototype service, where the joined data could be accessed via search interface. The actual effort to implement this service was minimal, which was very promising for the implementation of future tools and services.

The overall objective of the presented approach is to limit the effort for implementing new research tools, while maximizing re-use of data and code. The desired effects of data unification are conservation, interlinkage and unification of the data access. Additionally, there is the long term effort to unify the underlying code base, so that implementing a new tool could be just a matter of selecting

existing software components. The complete application to join various data silos, as described in this work, goes through the following phases:

1. Recording the characteristics and characteristic values of the research projects.
2. Creating the CMDI profiles.
3. Transfer the data into the component registry.
4. Prototypical implementation of a tool to show that the profiles are complete and correct.
5. Conversion or re-implementation of research projects via component registry.

Realizing described middleware is a work for years. Our institute's middleware is still being developed and CMDI representations only cover the most commonly used features. Individual features, better duplication detection, noting the source for bits of information, etc. have to be added in the future. In the short term, it was not feasible to recreate old projects using the newly created data repository. First, users would not benefit from this change and second, the newly created interconnection between the resources offer much more possibilities that the projects interfaces needed a complete overhaul.

Tools of the Digital Humanities have shown to be successful in supporting research on books. For instance, our institute provides several tools for doing research on textbooks and curricula. Unfortunately, not all institutes have the possibilities and the qualified staff to set up their own Digital Humanities architecture. Hence, in the future, we will enhanced our repository to be ready to cover additional data coming from other educational media research projects, from all around the world.

**Author Contributions:** Software, C.S.; Writing—original draft, F.F. and C.S.; Writing—review and editing, E.W.D.L. All authors have read and agreed to the published version of the manuscript.

**Funding:** This research received no external funding.

**Conflicts of Interest:** The authors declare no conflict of interest.

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
