# Peer review of "Visualization, Interaction and Analysis of Heterogeneous Textbook Resources"

_futureinternet, doi:10.3390/fi12100176_

Round 1

Reviewer 1 Report

Summary:
The paper aims to address the problem of data silos that results from research projects conducted by researchers who have limited understanding of data sustainability, data reusability and standards. Authors present a CMDI (Component Metadata Infrastructure) based approach to implement a repository for research project to add data is retroactively.

Merits of the Paper:
Research data silos is a new and the solution provided is innovative and uses a combination of techniques from different disciplines to address research data silos problem. Also, the work done to idenity characteristics and Characteristics values, as well as creating CMDI profiles and transfer data to registry is considerable contribution to the area. Paper has a logical structure and the article is correctly written. Method is clearly expalined and demonstrated.

Issues:
- Title of the paper maybe a bit broader than the content speciality.
- I suggest that you site all the project resources listed for recording characteristics and characteristic Values. Some people may not be familiar with one or more.

- Some cited sources show as question mark in text. I am not sure if this is a citing tool bug or authors forgot to pull the source?
- Figures need to show in the paper after refering to them in text (not before).
-Also, figure are not always well-proportioned and aesthetically placed in the paper (consistancy issue).

Recommendation:
I reommend accepting paper after addressing the minor issues.

Author Response

Thank you for your time used in reading our paper and in giving your precious insights.

Point 1: Title of the paper maybe a bit broader than the content speciality. 

Response 1:

Point 2: I suggest that you site all the project resources listed for recording characteristics and characteristic Values. Some people may not be familiar with one or more.

Response 2: site all the project resources listed on page 3

Point 3: Some cited sources show as question mark in text. I am not sure if this is a citing tool bug or authors forgot to pull the source?

Response 3 solved the cited sources problem on pages 1 and 5.

Point 4: Figures need to show in the paper after refering to them in text (not before).

Response 4: now the figures are shown after they are cited in the paper.

Point 5: Also, figures are not always well-proportioned and aesthetically placed in the paper (consistancy issue).

Response 5: figures are not well-proportioned and aesthetically placed in the paper

Reviewer 2 Report

The document is very interesting. The focus of research is middleware. It would be better to fully explain the effect of this component even if it is early in the development phase.

Author Response

Thank you for your time used in reading our paper and in giving your precious insights.

Point 1: It would be better to fully explain the effect of this component even if it is early in the development phase.

Response 1: we insert explanations of the effect of this component in the conclusion on page 13